# Erase to Enhance: Data-Efficient Machine Unlearning in MRI Reconstruction

**Yuyang Xue**[1]                                                        YUYANG.XUE@ED.AC.UK
**Jingshuai Liu**[1]                                                          JLIU11@ED.AC.UK
**Steven McDonagh**[1]                                              S.MCDONAGH@ED.AC.UK
**Sotirios A. Tsaftaris**[1]                                           S.TSAFTARIS@ED.AC.UK
[1] *School of Engineering, The University of Edinburgh, Edinburgh, EH9 3FG, UK*

**Editors:** Accepted for publication at MIDL 2024

## Abstract

Machine unlearning is a promising paradigm for removing unwanted data samples from a trained model, towards ensuring compliance with privacy regulations and limiting harmful biases. Although unlearning has been shown in, e.g., classification and recommendation systems, its potential in medical image-to-image translation, specifically in image reconstruction, has not been thoroughly investigated. This paper shows that machine unlearning is possible in MRI tasks and has the potential to be of benefit for bias removal. We set up a protocol to study how much shared knowledge exists between datasets of different organs, allowing us to effectively quantify the effect of unlearning. Our study reveals that combining training data can lead to hallucinations and reduced image quality in the reconstructed data. We use unlearning to remove hallucinations as a proxy exemplar for the removal of undesirable data. We show that machine unlearning is possible without full retraining. Furthermore, our observations indicate that maintaining high performance is feasible even when using only a subset of retain data. We have made our code publicly available.
**Keywords:** Machine Unlearning, MRI Reconstruction, De-biasing

## 1. Introduction

As the dependence on data-driven learning grows, model generalisation remains a key challenge. Common strategies include the aggregation of multiple data sets to expand the available data distributions, as a means of improving generalisation (Teney et al., 2021). However, such data agglomeration raises concerns about privacy, perpetuation of biases, and, in the case of image reconstruction, may generate hallucinations. In particular, successful magnetic resonance imaging (MRI) reconstruction of distinct organ types, under a single trained model, requires generalisation abilities. We corroborate recent work and find that model generalisation can be strengthened using data aggregation (Sect. 4.1). However, such aggregation leads to hallucinations that manifest themselves as false structures and artifacts in the reconstructed images (Darestani et al., 2021). Crucially, such resulting effects harm the clinical diagnostic value (Zbontar et al., 2018). Figure 1 shows an example of how learning from combined brain and knee data training can result in unwanted hallucinations in structures such as the corpus callosum, which could potentially lead to misdiagnoses.

While machine unlearning has been shown to remove unwanted data and classes in other tasks (e.g., classification), research on the integration of machine unlearning into image reconstruction remains, at best, nascent (Nguyen et al., 2022).

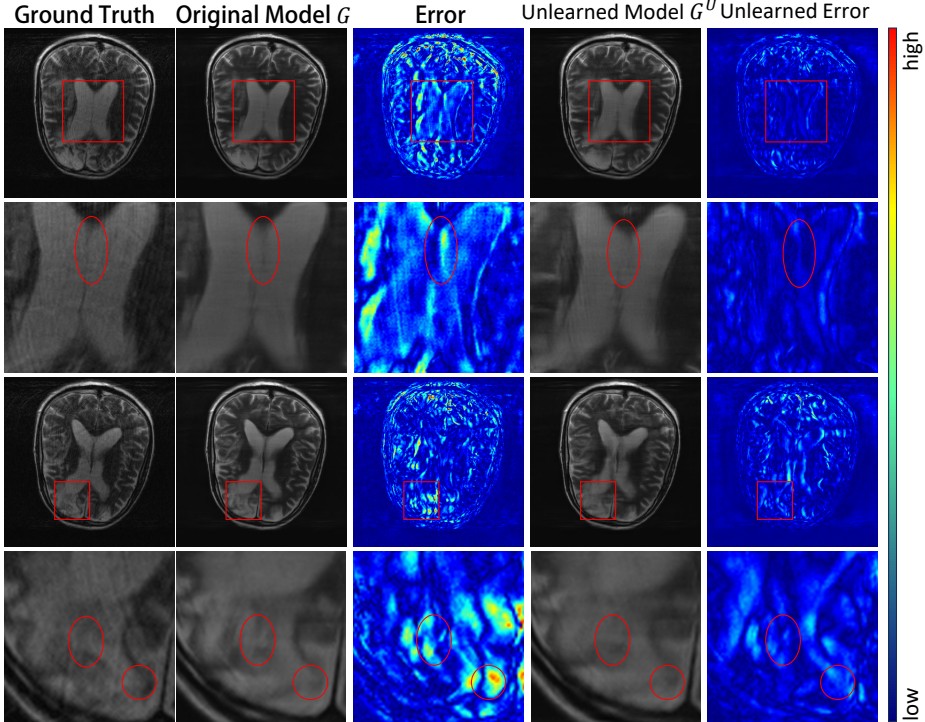

Figure 1: The original model $G$ trained with combined brain and knee data shows halluci­nations (red circles). The unlearned model $G^U$ can remove artifacts originating from such an anatomy shift, reducing the overall reconstruction error.

This paper sets up a protocol to analyse what models can learn about datasets from different distributions and uses this protocol to study machine unlearning. The protocol con­sists of training MRI reconstruction models with data combined from different anatomies, as obtained with multiple coils, which we have found can produce unwanted artefacts (see Fig­ure 1). We find the "knowledge gap" between the two datasets to effectively quantify this anatomy shift. We then propose suitable unlearning models that can remove such a shift. Thus, we use this setup as a proxy exemplar for unwanted data removal[1]. We highlight the task difficulty: existing well-defined settings of unlearning focus on classification and whole class removal. Instead, our task not only deals with a generative image-to-image task but also requires the effective removal of misgenerated structure whilst accurately preserving the anatomical features of a patient's body. To the best of our knowledge, we are the first to establish unlearning algorithms for a reconstruction task c.f. previously considered clas­sification tasks. Our contributions can be summarised as follows: (1) We propose a formal problem formulation and definition of machine unlearning within the MRI reconstruction

---

1. Ideally, we would consider patient data with unique characteristics, e.g., patients with titanium im­plants, abnormal anatomy, etc. Such patient data would have been ideal for studying unlearning but unfortunately open datasets with real $k$-space data are lacking.

domain. (2) We adapt machine unlearning approaches to MRI reconstruction and verify that successful machine unlearning allows for effective artifact removal and improves reconstruction quality. (3) In contrast to existing approaches, which necessitate access to an entire dataset, we demonstrate that unlearning algorithms can operate effectively with only a restricted fraction of data and achieve data-efficient unlearning for MRI reconstruction.

## 2. Related Works

**MRI Reconstruction** reconstructs medical images from acquired Fourier domain samples, known as the $k$-space. Deep learning reconstruction enables high-quality image reconstruction from undersampled $k$-space data. This is achieved by training networks to capture complex patterns and features within MRI data (Schlemper et al., 2017), learning mappings between the under- and fully sampled $k$-space (Akçakaya et al., 2019), or directly in the image domain (Wang et al., 2016). However, recent research shows that such models produce artifacts, overlook small structural changes, and have a decrease in performance under distribution shifts (Antun et al., 2020; Darestani et al., 2021). Inspired by these challenges, we set up a proxy exemplar of data removal.

**Machine Unlearning** aims to address the need for data privacy and regulatory compliance (Liu and Tsaftaris, 2020; Xu et al., 2023). It is the process of removing the influence of a set of data from a trained model, effectively "forgetting" this set without retraining the model from scratch (Ginart et al., 2019; Su et al., 2022). Numerous studies have emerged using data-based (Bhadra et al., 2021; Tarun et al., 2023; Graves et al., 2021) or model-based algorithms (Neel et al., 2021; Chourasia and Shah, 2023; Jia et al., 2023). However, these approaches are mainly suited to classification tasks. Parallel to this work, an image-to-image machine learning framework for generative models was developed Li et al. (2024). The idea is simple and borrows from a class of unlearning algorithms for classification tasks (see noisy labelling approaches discussed in Section 4): they introduce random noise during image generation (of the diffusion model) to degrade the model performance on the forget data. Similar to many unlearning methods, it relies on having access to the full original database, which can be unrealistic in medical practice.

**Machine Unlearning in Medical Imaging** Hartley et al. (2023) showed that networks could pose a risk of retaining and potentially revealing the distinctive characteristics of patients. Liu and Tsaftaris (2020) used the Kolmogorov-Smirnov distance to detect whether a model has used/forgotten a query dataset. Su et al. (2022) explored patient-wise forgetting and revealed that forgetting medical image data from a patient is harder than other vision tasks. These works highlight several challenges in medical imaging settings: models can easily learn unwanted relationships, and there is considerable similarity between medical data. While they don't present algorithms for image reconstruction, their findings help enforce why a suitable proxy exemplar is required,[2] why unlearning in image-to-image tasks is hard and why a delicate balance of unlearning and performance is required. Furthermore, as fully-sampled $k$-space data are extremely large and rare, unlearning methods that can relax the need for complete access to all training data should be explored.

---

2. We found that unlearning a "single patient's $k$-space data" is impossible to effectively quantify, it would not guarantee creating a significant performance gap to be observed.

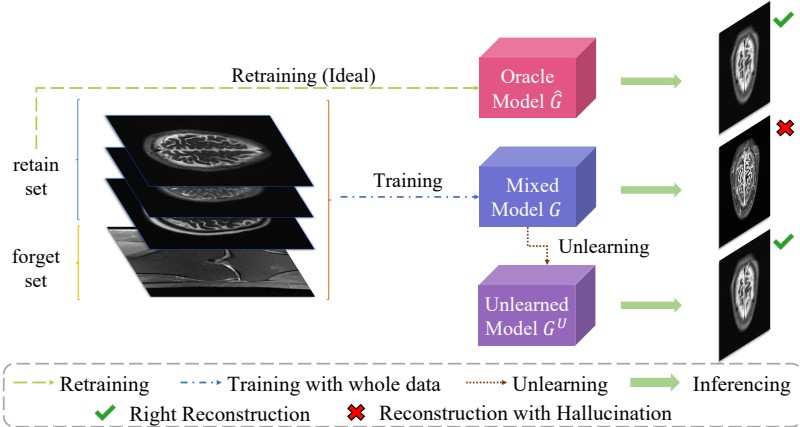

Figure 2: Machine unlearning in MRI reconstruction overview. The oracle model and the original model are trained on retain set and composite set (retain + forget), respectively. Taking advantage of the original model by employing an unlearning algorithm can quickly adapt to data removal requests instead of retraining.

## 3. Methodology

**Our main goal** is to effectively investigate machine unlearning for MRI reconstruction. We set up a proxy exemplar where the task is to reduce artefacts and hallucinations without reducing diagnostic precision. Utilising unlearning algorithms, we systematically erase the influence of a forget dataset from the model by evaluating image performance. We evaluate the integrity of reconstructed images and found that the performance of the models remains uncompromised after unlearning. Lastly, we demonstrate the practical implementation of our methodology on MRI data that limits the size of the training set, highlighting its potential for real-world application. The overview of the pipeline is shown in Figure 2.

### 3.1. Definitions of Unlearning in the Context of Reconstruction Tasks

We address the scenario where a reconstruction model $G$, parameterised by $\theta$, is initially trained on a combined dataset that comprises both a retain set $\mathcal{D}_r$ and a forget set $\mathcal{D}_f$, optimised via the loss function $\mathcal{J}(\theta)$. We posit the existence of an ideal oracle model $\hat{G}$, exclusively trained on $\mathcal{D}_r$ and thus uncontaminated by $\mathcal{D}_f$. The objective of the unlearning algorithm $\mathcal{A}$, applied to $G$, is twofold: (1) *to preserve or enhance the performance of the model in the retain set distribution $\mathcal{D}_r$*, ideally achieving the upper performance bound of $\hat{G}$ and (2) based on the premise of not destroying structural integrity, *to eliminate the impact of $\mathcal{D}_f$ as much as possible*, that is, to reduce the performance in the forget distribution. We call the unlearned model $G^U$. In practical applications, accessibility to the training dataset is usually restricted due to privacy or large storage requirements, the oracle model $\hat{G}$ cannot be retrained, forcing the unlearning algorithm $\mathcal{A}$ to operate on $G$ with the $\mathcal{D}_f$ alone and with limited access to a subset of retain data $\mathcal{D}_r'$. We selected two test sets to measure the effect of unlearning, namely $\mathcal{D}_r^t$ of the retain distribution and $\mathcal{D}_f^t$ of the forget distribution.

### 3.2. Protocol Setup

We present a proof-of-concept protocol for machine unlearning for MRI reconstruction. We treat the brain as retain data $\mathcal{D}_r$ and the knee as forget data $\mathcal{D}_f$. An oracle model $\hat{G}$ trained exclusively on multi-coil brain MRI data $\mathcal{D}_r$ establishes the gold standard for evaluating performance for unlearning algorithms. To simulate a real-world scenario, we then train the original model $G$ on a combined dataset comprising data from $\mathcal{D}_r$ and $\mathcal{D}_f$, each of which has a different anatomy. Subsequent application of unlearning algorithms removes the influence of knee data incorporated during the original training phase.

There is no precedent in evaluating the quality of forgetting in MRI reconstruction; therefore, we employ image quality assessment metrics as indicators of successful unlearning. An increase in performance on $\mathcal{D}_r$, accompanied by a decrease on $\mathcal{D}_f$, suggests that the model is redirecting its attention towards $\mathcal{D}_r$, the desired in-distribution data. Compared to pre-unlearning performance, this shift in the model manifests itself through improved image quality both quantitatively and qualitatively. To evaluate the efficacy of the unlearning process, we performed comparative analyses using several baselines and unlearning methods. The goal is to discern which algorithm achieves a balance of efficient unlearning while maintaining, or ideally enhancing, the model performance on $\mathcal{D}_r^t$. Since the unlearning process should be acute and accurate, rather than retraining the entire model, we restrict the unlearning steps to 10% of the training epochs. This proof-of-concept underscores the potential of machine unlearning to address bias removal in AI models, and serves as a basis for future research in deploying machine unlearning within medical imaging domains.

### 3.3. Machine Unlearning Methods for Reconstruction

We introduce machine unlearning into MRI reconstruction, adapting three major techniques, originally developed for classification tasks, to suit an image-to-image reconstruction context. All algorithmic details are found in the Appendix.

**Fine-tuning (FT)** is the most widely deployed unlearning strategy, often used to address distribution shifts and adapt models to new distributions. In machine unlearning, the fine-tuning process is only implemented on $\mathcal{D}_r$, using a loss function $\mathcal{J}(\theta, \mathbf{x}, \mathbf{y})$, as

$$\mathcal{L}_{FT} = \mathcal{J}(\theta, \mathbf{x}_r, \mathbf{y}), \ \mathbf{x}_r \in \mathcal{D}_r, \tag{1}$$

where $\mathcal{J}(\theta, \mathbf{x}, \mathbf{y})$ denotes the reconstruction loss, normally $\mathcal{L}_1$ between the undersampled input $\mathbf{x}$ and the ground truth $\mathbf{y}$, and $\theta$ refers to the model parameters. We use the same notation for the methods in the following. However, as fine-tuning does not impose any penalty on the forget data, the unlearning tends to be less efficient than other methods.

**Gradient Ascent (GA)** maximises the training loss w.r.t. forget samples, effectively decreasing model accuracy for knowledge removal (Halimi et al., 2022; Warnecke et al., 2021). However, without any restrictions, the model tends to forget all the knowledge obtained before unlearning. We hence add $\ell_1$-regularisation ($GA$-$\ell_1$) to constrain the weights to prevent divergence, which encourages sparsity in the model weights during training (Jia et al., 2023). Sparsity can lead to simpler and more interpretable models that may perform better when it comes to generalisation on unseen data by reducing overfitting. The regularised GA loss is defined by

$$\mathcal{L}_{GA_{\ell_1}} = -\mathcal{J}(\theta, \mathbf{x}_f, \mathbf{y}) + \gamma \|\theta\|_1, \ \mathbf{x}_f \in \mathcal{D}_f. \tag{2}$$

**Noisy Labelling (NL)** minimises the training loss by reassigning the label of forget samples by adding Gaussian noise with a hyperparameter $\lambda$ indicating the amount of noise, thus removing the mapping of the input to the OOD images (Graves et al., 2021; Gandikota et al., 2023). In MRI reconstruction, a false mapping to the target would lead to a considerable drop in global performance. We change the forget set labels as follows:

$$\mathcal{L}_{NL} = \mathcal{J}(\theta, \mathbf{x}_f, [\mathbf{y} + \lambda \mathcal{N}(0, I)]), \ \mathbf{x}_f \in \mathcal{D}_f. \tag{3}$$

## 4. Experiments

### 4.1. Dataset

We use FastMRI dataset (Knoll et al., 2020) to reconstruct multiple coil brain and knee images. We randomly selected T2-weighted brain volumes, yielding 5,898 slices in 1,000 volumes as the full retain dataset $\mathcal{D}_r$. The forget set $\mathcal{D}_f$ has 536 randomly selected knee image slices. We adhered to a retain-to-forget set ratio of 10:1. For validation, a separate subset was created by randomly choosing 1,198 slices. Two test sets are 100 volumes of brain and knee data, respectively. For data processing, an equispaced Cartesian sampling pattern was applied to simulate unacquired $k$-space lines. The acceleration rate and the centre fraction rate were set to 8-fold and 0.04, respectively.

### 4.2. Training Details

We selected the E2E-VarNet model (Sriram et al., 2020), which processes masked multi-coil $k$-space data as input, as the main network for reconstruction. The Sensitivity Map Estimation (SME) module inside computes sensitivity maps from the input $k$-space, together with a sequence of UNet cascades that iteratively refine outputs. We follow the training protocol of the original implementation, with 12 cascades, resulting in 29.9M total parameters. Models were trained using the Adam optimiser with a learning rate of 0.001 over 50 epochs without any data augmentation. We used the PyTorch Lightning framework on a NVIDIA A100 Tensor Core GPU. Image quality was evaluated using peak signal-to-noise ratio (PSNR) and structural similarity index (SSIM) (Wang et al., 2004) as metrics.

### 4.3. Unlearning Instantiating and Results

We analysed the experiments in three dimensions (Jia et al., 2023): accuracy on brain test data $\mathcal{D}_r^t$ as brain test accuracy (BTA), aiming to demonstrate the capacity to retain the learned target knowledge, accuracy on knee test data $\mathcal{D}_f^t$, denoted by knee test accuracy (KTA) and serving as a metric to gauge the efficacy of unlearning algorithms, and additionally accuracy on forget data $\mathcal{D}_f$ as unlearning accuracy (UA) which is used to confirm the unlearning quality when combining it with KTA. We also added run-time efficiency (RTE) as another metric for the evaluation of the unlearning algorithm. Performances are highlighted in Table 1 and Figure 3 shows a radar chart of comparison of all algorithms.

Table 1: Quantitative evaluation of unlearning methods. All models were tested on brain test accuracy (BTA), knee test accuracy (KTA), and unlearning accuracy (UA). An ideal unlearning outcome should be **as close as possible** to the performance of oracle model $\hat{G}$, highlighted in bold text. $\lambda$ is set to $10^{-5}$ for *NL* and *NL-FT*. Except for *Full FT*, unlearning methods were implemented with 10% retain data.

| **Method** | Brain $\mathcal{D}_r^t$ (BTA) | | Knee $\mathcal{D}_f^t$ (KTA) | | Unlearning $\mathcal{D}_f$ (UA) | |
|---|---|---|---|---|---|---|
| | **PSNR** | **SSIM** | **PSNR** | **SSIM** | **PSNR** | **SSIM** |
| $G$ | 39.07±2.03 | 0.964±0.018 | 36.65±2.44 | 0.894±0.059 | 37.13±2.19 | 0.905±0.045 |
| $\hat{G}$ | **39.36**±2.04 | **0.965**±0.017 | **31.82**±2.76 | **0.834**±0.065 | **31.74**±2.12 | **0.892**±0.052 |
| *Fine-tuning* | | | | | | |
| *FT* | **39.18**±2.16 | **0.965**±0.019 | 36.67±2.39 | 0.893±0.059 | 37.18±2.18 | 0.905±0.045 |
| *Full FT* | 39.12±2.02 | **0.965**±0.017 | 36.26±2.34 | 0.889±0.059 | 36.60±2.07 | 0.901±0.046 |
| *Unlearning Algorithms* | | | | | | |
| $GA$-$\ell_1$ | 16.19±2.38 | 0.172±0.035 | 14.15±3.10 | 0.156±0.067 | 13.40±2.63 | 0.162±0.056 |
| *NL* | 16.44±1.58 | 0.502±0.035 | 17.39±2.89 | 0.421±0.089 | 17.04±2.28 | 0.428±0.072 |
| $GA$-$\ell_1$-*FT* | 31.81±2.09 | 0.910±0.025 | 27.09±2.49 | 0.735±0.074 | 26.90±1.57 | 0.739±0.025 |
| *NL-FT* | 38.24±1.94 | 0.956±0.019 | **31.18**±4.28 | **0.863**±0.068 | **31.72**±4.14 | **0.875**±0.053 |

**Oracle vs. Original** To show the domain gap existing between brain and knee data and hence quantify the influence of the forget set $\mathcal{D}_f$ on the reconstruction model, we compared the original model $G$, trained on both $\mathcal{D}_r$ and $\mathcal{D}_f$, and the oracle model $\hat{G}$, trained solely on brain data $\mathcal{D}_r$ to represent an upper bound for unlearning methods. We can see from Table 1 that these two models show differences in terms of evaluation metrics for each organ, with the oracle $\hat{G}$ better on brain data and worse on knee data. This is expected given the domain shift, but the gap of about 5dB on knee data implies the distribution dissimilarities between the two organs. The findings highlight the need for machine unlearning to counteract domain-related issues and maintain reliable image reconstruction quality.

**Fine-tuning does not unlearn** Fine-tuning aims to unlearn the forget set by shifting the learned data distribution towards the target domain. Two levels of fine-tuning are

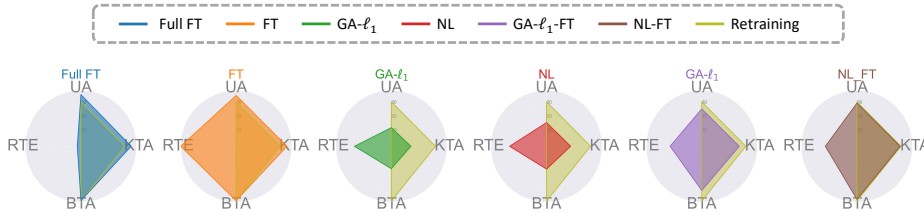

Figure 3: Unlearning approaches vs. oracle model $\hat{G}$. Unlearning accuracy (UA), brain test accuracy (BTA), and knee test accuracy (KTA) are shown in PSNR. The reciprocal of run-time efficiency (RTE) is normalised to $[0, 1]$ for ease of visualisation. *FT* and *NL-FT* achieve the best, closest to oracle with the highest RTE.

selected. One is to fine-tune $G$ on the whole $\mathcal{D}_r$, termed *Full FT*, which requires us to have access to large-scale training data which is impractical. The other is to fine-tune the model on a small subset of target data, $\mathcal{D}'_r$, termed *FT*. Both *Full FT* and *FT* marginally enhance the performance of the model on the target brain dataset. However, the ability to reconstruct knee images remains unaffected or, in the case of *FT*, even improved. This indicates that fine-tuning is insufficient in discarding knowledge of the forget set. These findings have inspired the integration of unlearning algorithms into the fine-tuning process.

**Combining unlearning and fine-tuning** We used $GA$-$\ell_1$ and *NL* as unlearning algorithms. $GA$-$\ell_1$ fine-tuned with $\mathcal{D}'_r$ is referred to as $GA$-$\ell_1$-*FT*, and the other is designated as *NL-FT*. We noted from Table 1 that *NL* and $GA$-$\ell_1$ lower the metrics for the knee test data, suggesting their effectiveness in the unlearning of domain knowledge. However, these methods also substantially deteriorated the reconstruction fidelity of brain images, indicating that unlearning algorithms, if applied solely with $\mathcal{D}_f$, can inadvertently compromise vital information in the target domain. Methods $GA$-$\ell_1$-*FT* and *NL-FT* combine fine-tuning to maintain the learned knowledge in the target domain during their unlearning steps, using the knowledge from $\mathcal{D}'_r$ to steer the unlearning direction toward the retain data distribution, as illustrated in Appendix Figure 4. According to the results in Table 1, both methods can effectively achieve model unlearning in a reasonable range, with *NL-FT*, which could be a promising unlearning method, showing a superior ability to maintain the reconstruction integrity compared to $GA$-$\ell_1$-*FT*. More details are provided in Appendix Table 2.

**Do we need all retain data?** We evaluate whether having full access to the retain dataset results in a more effective unlearning compared to having limited access practically to a portion of the retain dataset. For these experiments we used a smaller retain dataset $\mathcal{D}'_r$, at percentages 1%, 5%, 10%, 20%, 50% of the full retain dataset $\mathcal{D}_r$. We evaluated performance considering two primary factors: run-time efficiency and PSNR of the brain test set $\mathcal{D}^t_r$. We expect that the more retain (in-distribution) data we used for fine-tuning, the better the results should be. Unexpectedly, a decrease in PSNR was observed when the model was fine-tuned using the complete $\mathcal{D}_r$. Figure 5 in the Appendix depicts the results on *NL-FT* and $GA$-$\ell_1$-*FT* using a second-degree polynomial fitting. It shows that, to some extent, as the amount of retained set data increases, the quality of image reconstruction does not improve, but rather deteriorates. More information is in Appendix Table 3.

## 5. Conclusion

Our study is the first to investigate the application of machine unlearning to MRI reconstruction, marking significant progress in medical imaging research with respect to artifacts reduction without comprehensive retraining. Our findings underscore the potential of unlearning approaches to improve image quality and preserve data integrity by fine-tuning in a data-efficient way. Further investigation is necessary to understand the decrease in using larger retain datasets and to determine the optimal subset size for successful unlearning. This work lays the groundwork for future research to extend these methods to more complex datasets and enhance their efficiency in clinical settings, facilitating accuracy for clinical decision-making and the ethical standards demanded by society.

## Acknowledgments

Y. Xue thanks additional financial support from the School of Engineering, the University of Edinburgh. S.A. Tsaftaris also acknowledges the support of Canon Medical and the Royal Academy of Engineering and the Research Chairs and Senior Research Fellowships scheme (grant RCSRF1819\8\25), of the UK's Engineering and Physical Sciences Research Council (EPSRC) (grant EP/X017680/1) and the National Institutes of Health (NIH) grant 7R01HL148788-03. We also thank Jinghan Sun for the help.

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

# Appendix A. Unlearning Algorithms

We list the algorithm for the unlearning algorithm adaptation for $GA$-$\ell_1$-$FT$ and $NL$-$FT$. Figure 4 gives a illustration of regarding the issue of using fine-tuning to effectively suppress gradient ascent and cause the model to diverge and collapse.

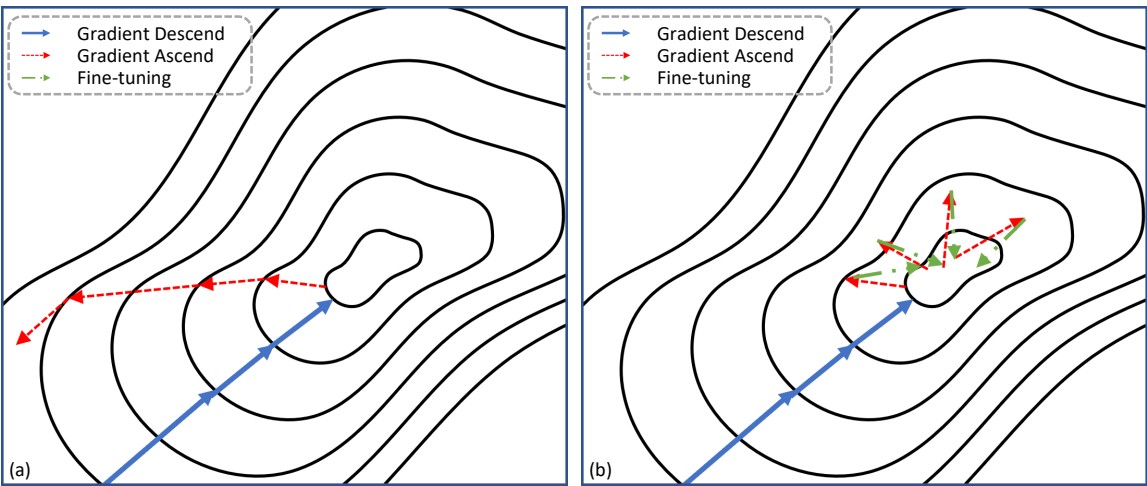

Figure 4: The gradient ascent ($GA$-$\ell_1$-$FT$) and gradient ascent with fine-tuning ($NL$-$FT$).

---

**Algorithm 1:** Gradient Ascent with $l_1$-regularisation Unlearning with Fine-tuning

---

Samples: $\mathbf{x}_i, \mathbf{y}_i \in \mathcal{D}_f, \hat{\mathbf{x}}_j, \hat{\mathbf{y}}_j \in \mathcal{D}_r$
Network parameters: $\theta$
**for** $\mathbf{x}_i \in \mathcal{D}_f, \bar{\mathbf{x}}_j \in \mathcal{D}_r$ **do**
$\quad L = -\mathcal{L}_{GA_{\ell_1}}(\mathbf{x}_i, \mathbf{y}_i) + L_{FT}(\hat{\mathbf{x}}_j, \hat{\mathbf{y}}_j)$
$\quad \theta \leftarrow \theta - \eta\nabla_\theta L$
**end**

---

**Algorithm 2:** Noisy Labelling Unlearning with Fine-tuning

---

Samples: $\mathbf{x}_i \in \mathcal{D}_f, \hat{\mathbf{x}}_j, \hat{\mathbf{y}}_j \in \mathcal{D}_r$
Network parameters: $\theta$
Hyperparameter: $\lambda$
**for** $\mathbf{x}_i \in \mathcal{D}_f, \bar{\mathbf{x}}_j \in \mathcal{D}_r$ **do**
$\quad L = \mathcal{L}_{NL}(\mathbf{x}_i, \mathbf{y}_i + \lambda\mathcal{N}(0, I)) + L_{FT}(\hat{\mathbf{x}}_j, \hat{\mathbf{y}}_j)$
$\quad \theta \leftarrow \theta - \eta\nabla_\theta L$
**end**

---

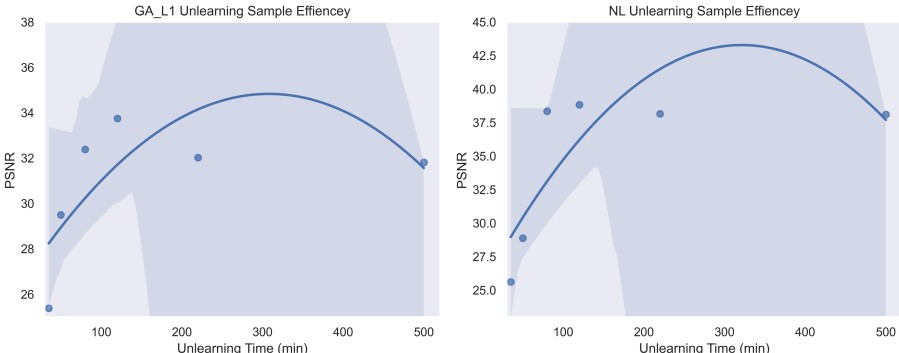

Figure 5: The Pareto optimum to achieve high unlearning efficiency may be found in the fitting curve of the BTA to unlearning time, which is directly related to retain sample usage.

## Appendix B. Unlearning Experiment Results in Details

We give detailed results for all the related experiments we conducted. Table 2 recorded the extra 5 epochs of unlearning and we take the last checkpoint's performance to show on Table 1. The ablation study of the proportions of retain data set we shall use during our unlearning algorithm for fine-tuning is shown in Table 3.

Table 2: Detailed test results for different unlearning algorithms, shows the unlearning dynamics as well.

| Method | Epochs | Brain $\mathcal{D}_r^t$ (KTA) | | Knee $\mathcal{D}_f^t$ (KTA) | |
|---|---|---|---|---|---|
| | | **PSNR** | **SSIM** | **PSNR** | **SSIM** |
| $G$ | 50 | 39.07±2.03 | 0.964±0.018 | 36.65±2.44 | 0.894±0.059 |
| $\hat{G}$ | 50 | **39.36**±2.04 | **0.965**±0.017 | **31.82**±2.76 | **0.834**±0.065 |
| *Full FT* | 51 | 39.080±2.01 | 0.9640±0.018 | 36.624±2.40 | 0.8931±0.059 |
| | 52 | 39.008±2.00 | 0.9646±0.017 | 36.406±2.39 | 0.8915±0.059 |
| | 53 | 39.024±2.01 | 0.9649±0.018 | 36.173±2.34 | 0.8898±0.059 |
| | 54 | 39.207±2.01 | 0.9651±0.017 | 36.267±2.38 | 0.8905±0.059 |
| | 55 | 39.119±2.02 | 0.9651±0.017 | 36.260±2.34 | 0.8894±0.059 |
| *FT* | 51 | 39.278±2.03 | 0.9651±0.017 | 36.756±2.40 | 0.8942±0.059 |
| | 52 | 39.140±2.02 | 0.9647±0.017 | 36.715±2.41 | 0.8938±0.059 |
| | 53 | 39.181+2.02 | 0.9649±0.017 | 36.706±2.41 | 0.8938±0.059 |
| | 54 | 39.224±2.03 | 0.9650±0.017 | 36.718±2.40 | 0.8935±0.059 |
| | 55 | 39.179±2.16 | 0.9647±0.019 | 36.672±2.39 | 0.8936±0.059 |
| *GA-$\ell_1$* | 51 | 0.000±2.03 | 0.0019±0.001 | 0.000±2.91 | 0.0017±0.002 |
| | 52 | 0.000±2.42 | 0.0000±0.001 | 0.000±2.71 | 0.0000±0.001 |
| | 53 | 1.272±1.27 | 0.0073±0.007 | 0.007±2.77 | 0.0037±0.004 |
| | 54 | 2.357±2.36 | 0.0080±0.008 | 2.314±2.88 | 0.0069±0.006 |
| | 55 | 16.186±2.38 | 0.1724±0.035 | 14.148±3.10 | 0.1561±0.067 |
| *NL* | 51 | 16.478±1.56 | 0.5026±0.034 | 17.436±2.90 | 0.4135±0.083 |
| | 52 | 16.442±1.56 | 0.4970±0.032 | 17.469±2.92 | 0.4189±0.090 |
| | 53 | 16.458±1.57 | 0.5173±0.035 | 17.430±2.92 | 0.4174±0.092 |
| | 54 | 16.472±1.58 | 0.5003±0.035 | 17.472±2.92 | 0.4217±0.094 |
| | 55 | 16.444±1.58 | 0.5015±0.035 | 17.385±2.89 | 0.8928±0.089 |
| *GA-$\ell_1$-FT* | 51 | 30.626±1.95 | 0.9044±0.025 | 27.807±2.50 | 0.7546±0.070 |
| | 52 | 31.546±2.01 | 0.907±0.024 | 27.006±2.52 | 0.7403±0.070 |
| | 53 | 29.570±1.94 | 0.8632±0.033 | 27.952±2.55 | 0.7534±0.069 |
| | 54 | 31.744±2.00 | 0.9106±0.024 | 26.811±2.54 | 0.7247±0.072 |
| | 55 | 31.813±2.09 | 0.9106±0.025 | 27.094±2.49 | 0.7353±0.074 |
| *NL-FT* | 51 | 37.592±1.98 | 0.9603±0.020 | 30.168±4.68 | 0.8514±0.071 |
| | 52 | 37.732±1.92 | 0.9612±0.019 | 30.365±4.47 | 0.8568±0.070 |
| | 53 | 37.953±1.93 | 0.9616±0.019 | 30.503±4.58 | 0.8573±0.069 |
| | 54 | 37.519±1.98 | 0.9596±0.018 | 30.595±4.42 | 0.8605±0.070 |
| | 55 | 38.243±1.94 | 0.9565±0.019 | 31.185±4.28 | 0.8630±0.068 |

Table 3: Detailed results for unlearning with various retain data $\mathcal{D}'_r$ proportions.

| Size | Methods | Brain $\mathcal{D}^t_r$ (BTA) | | Knee $\mathcal{D}^t_f$ (KTA) | |
|------|---------|------|------|------|------|
| | | **PSNR** | **SSIM** | **PSNR** | **SSIM** |
| \ | $G$ | 39.07±2.03 | 0.964±0.018 | 36.65±2.44 | 0.894±0.059 |
| | $\hat{G}$ | **39.36**±2.04 | **0.965**±0.017 | **31.82**±2.76 | **0.834**±0.065 |
| 1% | $FT$ | 39.182±2.02 | 0.9649±0.018 | 36.743±2.40 | 0.8942±0.059 |
| | $NL$-$FT$ | 25.607±2.53 | 0.8800±0.027 | 31.167±4.96 | 0.4202±0.096 |
| | $GA$-$\ell_1$-$FT$ | 25.385±1.38 | 0.7226±0.040 | 26.824±2.69 | 0.6922±0.068 |
| 5% | $FT$ | 39.051±2.00 | 0.9614±0.018 | 36.728±2.50 | 0.8942±0.061 |
| | $NL$-$FT$ | 28.877±2.78 | 0.9142±0.028 | 31.297±4.79 | 0.4491±0.078 |
| | $GA$-$\ell_1$-$FT$ | 29.499±1.58 | 0.8440±0.025 | 25.924±2.46 | 0.6987±0.068 |
| 10% | $FT$ | 39.089±2.03 | 0.9621±0.019 | 35.452±2.64 | 0.8860±0.062 |
| | $NL$-$FT$ | 38.243±1.94 | 0.9565±0.019 | 31.185±4.28 | 0.8630±0.068 |
| | $GA$-$\ell_1$-$FT$ | 32.128±1.87 | 0.9179±0.022 | 30.173±2.44 | 0.8045±0.061 |
| 20% | $FT$ | 39.070±2.01 | 0.9628±0.019 | 35.147±2.48 | 0.8845±0.061 |
| | $NL$-$FT$ | 38.828±2.05 | 0.9614±0.020 | 31.468±4.83 | 0.4577±0.085 |
| | $GA$-$\ell_1$-$FT$ | 33.746±1.96 | 0.9348±0.023 | 30.150±2.40 | 0.8085±0.062 |
| 50% | $FT$ | 39.048±1.99 | 0.9625±0.017 | 35.115±2.49 | 0.8844±0.061 |
| | $NL$-$FT$ | 38.484±1.98 | 0.9597±0.019 | 31.152±4.78 | 0.4591±0.078 |
| | $GA$-$\ell_1$-$FT$ | 32.381±1.80 | 0.9193±0.022 | 29.841±2.48 | 0.7922±0.069 |
| 100% | $FT$ | 39.119±2.02 | 0.9651±0.017 | 36.260±2.34 | 0.8894±0.059 |
| | $NL$-$FT$ | 37.464±1.88 | 0.9587±0.019 | 31.122±4.88 | 0.4586±0.077 |
| | $GA$-$\ell_1$-$FT$ | 31.928±1.65 | 0.9179±0.021 | 29.461±2.55 | 0.8021±0.065 |

