# OpenReview forum: "Erase to Enhance: Data-Efficient Machine Unlearning in MRI Reconstruction"
_MIDL.io/2024/Conference — MIDL 2024 Poster_

### Official Review · Reviewer_tpqe · 2024-02-27

**Confidence:** 4
**Preliminary Rating:** 4
**Recommendation:** Poster
**Final Rating:** 4

**Summary:**

The manuscript „Erase to Enhance: Data-Efficient Machine Unlearning in MRI Reconstruction “ describes a modern and very innovative approach of image reconstruction in magnetic resonance imaging (MRI).
Objects of this approach are twofold: 1) An ideally identical performing reconstruction model for the wanted data (retain set D_r, here brain data) and 2) an efficient untraining method such that the model G^U eliminates as much as possible falsely structured introduced by through presence of the unwanted data (forget set D_f, here knee data) during the training. In addition to the direct benefits of unlearning a positive side effect is training data reduction, i.e. limiting the dataset size of large k-space data for training, is pointed out. Overall, the manuscript is in most parts well written and scientifically sound to show a good understanding of the topic. The main reasoning why untraining should be done in contrast to other methods such as fine-tuning is not very strong backed by the findings in this work.

**Strengths:**

Very sound description of certain challenges in MRI reconstruction tasks. The authors show with the high quality of references to literature that they monitor past and recent developments in the field and put their work into good relation to published work.

**Weaknesses:**

Some part could benefit from clearer writing, e.g. in 4.2 as in Sriram 2020 your NN ended with k-space data, didn't it? Please point out, that you use FFT and sum-of-squares (or other) method to compute an image from k-space and combine multi coil data. Also, to emphasize that PSNR and SSIM are applied in image (and not k-) space. Please also explain, how you handle complex-valued k-space data.

**Detailed Comments:**

-	p.5 …we then train the original model G on a combined dataset…  suggesting to rename it to “…train the mixed model G on…”
-	suboptimal definitions of variables in Eq. 1 and 2  please use subscripts to indicate whether x is from the retain or forget set. In the end the mathematical expression should be clearly interpretable such that a user knows, how losses differ and what information they take into account.
-	4.2: Please clarify the handling of complex-valued k-space data.

**Justification Of Final Rating:**

The authors addressed my raised points and answered my questions. With that crucial points were clarified thanks. Overall, I will remain with my initial rating for weak accept to be presented at MIDL 2024.

**Justification Of The Preliminary Rating:**

Overall to me the manuscript is a valuable contribution to scientific discussion. It lacks of clarity and some details as I would expect in a full paper however, this is also sth. I do not necessarily expect for a conference contribution. The authors have done their experiments in a careful and well organized manner. The results results seem to me not as clear as the authors might state, which is totally okay. Still the approach is worth a scientific discussion.

**Questions To Address In The Rebuttal:**

-	Was benchmarking again conventional reconstruction methods (not a trained model) as gold standard done?
-	Couldn’t be the following alternative approach more explainable and straight forward:
Perform standard/low level image reconstruction, do image classification, exclude unwanted anatomies and then simply train on wanted and ‘good’ data the full reconstruction model?
-	Table 1: To me the Fine-tuning approach seems to result in the best overall metrics. Could you please comment on this? Why would I need unlearning then? In the paragraph below you refer to a Sub FT set. Is this identical to FT in Table 1? If so, I still can not completely follow you interpretation of the results which show – just by considering the overall quantitative metrics – that Fine-Tuning works best.

**Special Issue:**

No

---

> ### Author Response · Authors · 2024-03-15
> **Rebuttal to Reviewer tpqe (1)**
>
> The authors thank reviewer tpqe for the time and effort for reviewing our work and the constructive suggestions. Here is the rebuttal to the questions.
>
> ---
>
> **Q1**: *In 4.2 as in Sriram 2020 your NN ended with k-space data, didn't it? Please point out, that you use FFT and sum-of-squares (or other) method to compute an image from k-space and combine multi coil data. Also, to emphasize that PSNR and SSIM are applied in image (and not k-) space. Please also explain, how you handle complex-valued k-space data.*
>
> **A1**: Our model takes as input the masked $k$-space data, denoted as $\mathbf{k}_0$, which is originally in complex form and is converted into a two-channel real-valued representation. Using an end-to-end variational network (E2E-Varnet) backbone, the network refines the input to generate a reconstructed $k$-space representation, $\mathbf{k}$. Following the refinement process, we then convert the final k-space output back into image space. This conversion is performed by applying an inverse fast Fourier transform (IFFT), which transforms the k-space data into its corresponding spatial domain. To reconstruct the final image representation from the multicoil MRI data, we employ a root-sum-of-squares (RSS) combination method across the coil images for each pixel, yielding a single coherent image that reflects the combined signal.
>
> We wish to emphasise that all evaluation metrics, specifically `PSNR` and Structural Similarity Index Measure (`SSIM`), are computed on these reconstructed image-space data.
>
> ---
>
> **Q2**: *p.5 “…we then train the original model G on a combined dataset…” suggesting to rename it to “…train the mixed model G on…”*
>
> **A2**: While we appreciate the intention behind the suggestion to rename "the original model $G$" to "the mixed model $G$," we would like to explain our rationale for the terminology chosen within our text. The terms "original model," "unlearned model," and "oracle model" are consistently used in our manuscript as per the naming conventions established in notable previous works [1, 2] and guidelines from authoritative challenges such as the Google Unlearning Challenge [3].
>
> To further improve readability, we will add a footnote or parenthetical note at the first instance of "original model" on _page 5_, explaining that this term refers to the "mixed" state of $G$ when trained on a combined dataset of both in-distribution and out-of-distribution data.
>
> ---
>
> **Q3**: *Suboptimal definitions of variables in Eq. 1 and 2. please use subscripts to indicate whether x is from the retain or forget set. In the end the mathematical expression should be clearly interpretable such that a user knows, how losses differ and what information they take into account.*
>
> **A3**: Agreed. We update the manuscript and thank the reviewer for the suggestion.
>
> ---
>
> **Q4**: *Was benchmarking again conventional reconstruction methods (not a trained model) as gold standard done?*
>
> **A4**: Thanks for the comment. We later tested both multicoil brain and knee data in an acceleration rate of `8x` using `GRAPPA` to reconstruction and coil sensitivities were estimated using `ESPIRiT`. Since the sampling pattern was pseudo-equispaced, multiple `GRAPPA` kernels were used to calculate the `GRAPPA` images. The final `SSIM` is `0.844` for brain images and `0.693` for knee images in average, which is similar but inferior to the oracle model $\hat{G}$. We omitted the conventional reconstruction method as we were doing *unlearning*, and comparing with a learning-based model is more meaningful in our context.
>
> ---
>
> To be continued on `Rebuttal to Reviewer tpqe (2)`.
>
> ---
>
> **Reference**:
>
> [1]. Liu, Y., Ma, Z., Liu, X., Liu, J., Jiang, Z., Ma, J., Yu, P., Ren, K., 2021. Learn to Forget: Machine Unlearning via Neuron Masking.
>
> [2]. Kurmanji, M., Triantafillou, P., Hayes, J., Triantafillou, E., 2023. Towards Unbounded Machine Unlearning. Presented at the Thirty-seventh Conference on Neural Information Processing Systems.
>
> [3]. Google, 2023, NeurIPS 2023 Machine Unlearning Challenge. Available at: https://unlearning-challenge.github.io/assets/data/Machine_Unlearning_Metric.pdf (Accessed: 15 March 2024).

---

> ### Author Response · Authors · 2024-03-15
> **Rebuttal to Reviewer tpqe (2)**
>
> Continued from `Rebuttal to Reviewer tpqe (1)`.
>
> ---
>
> **Q5**: *Couldn’t be the following alternative approach more explainable and straight forward: Perform standard/low level image reconstruction, do image classification, exclude unwanted anatomies and then simply train on wanted and ‘good’ data the full reconstruction model?*
>
> **A5**: While the suggestion is indeed a valid strategy under certain circumstances, we would like to underscore that it addresses a different problem setting from the one our paper targets. Our work is rooted in the concept of machine unlearning, which is particularly relevant to cases where a machine learning model has already been deployed and trained on a dataset that includes both desired and undesired data. The key challenge we address is the ability to remove the influence of unwanted data from this pre-trained model without retraining it from scratch, a scenario that is becoming increasingly common in practice due to constraints related to time, computational resources, and accessibility of the original training data, etc.
>
> ---
>
> **Q6**: *Table 1: To me the Fine-tuning approach seems to result in the best overall metrics. Could you please comment on this? Why would I need unlearning then? In the paragraph below you refer to a Sub FT set. Is this identical to FT in Table 1? If so, I still can not completely follow you interpretation of the results which show – just by considering the overall quantitative metrics – that Fine-Tuning works best.*
>
> **A6**: Thank you for pointing out this. As other reviewer also pointed out this, we here apologise for the misunderstanding.  The inconsistency of `Sub_FT` and `FT` will be amended in the manuscript to ensure clarity and uniformity. The term `Sub-FT` mentioned in _Section 4.3_ is intended to correspond to `FT` as in _Table 1_. Similarly, `Full-FT` in both the text and the table refers to the same fine-tuning method applied across the full dataset. We will rectify this oversight to align the narrative of _Section 4.3_ and _Table 1_.

---

### Official Review · Reviewer_fLbn · 2024-02-28

**Confidence:** 3
**Preliminary Rating:** 3
**Final Rating:** 4

**Summary:**

The authors tackle the problem of unlearning, ie removing biases in the model from a subset of a model’s training data, in MRI.

To do so, the authors use the task of image reconstruction from k-space. Several unlearning methods are tested, as well as varying the amount of unlearning data. The authors report performance relative to an “unbiased” model.

**Strengths:**

The paper is well written and thorough. The task of unlearning is thoroughly introduced. Several unlearning techniques are presented and evaluated, as well as combinations of them. Additionally, the effect of varying the amount of unlearning data is asserted. Unlearning in MRI, for image generation, seems to be a novel problem.

**Weaknesses:**

Unfortunately, the task of unlearning in MRI **as presented** in this work feels artificial. Several potential applications are hinted at throughout the paper, such as removing artefacts ("hallucinations"), removing bias, or preventing patient data recovery. However, none of these applications are actually explored and the unlearning is presented as a goal in itself, which as far as the reviewer is aware, does not reflect a real life problem. The experiment considered, where a researcher would enhance a brain images dataset with knee images, does not represent a meaningful scenario. The considered evaluation metrics do not tell us if hallucinations are removed or if knee images can no longer be recovered, for example.

Moreover, the presented results are unconvincing and none seem to outperform the others. Most considered methods either provide no impact wrt their "oracle" network, or catastrophically degrade the results on the target "retain" dataset. Therefore, even though the problem is novel, there is seemingly no grounds for a potential future avenue of research.

**Detailed Comments:**

Minor criticisms:

Run-time efficiency (RTE) is introduced as a metric in section 4.3 but is never explained. Figure 3 displays it for some considered method but does not bring anything to a reader not familiar with the metric. Figure 3, with our without RTE, seems redundant with Table 1 and could be excluded from the article. Second 4.3 refers to Sub-FT and Full-FT yet table 1 considers FT and Full-FT. The results in Figure 4 seem very sparse and could do with more samples.

**Justification Of Final Rating:**

The authors have thoroughly responded to issues raised in the initial review. The motivation behind the problem as posed and the datasets used should be obvious enough for a new reader. The performances of the proposed methods lay the groundwork for future research.

**Justification Of The Preliminary Rating:**

While I am very confident in my technical understanding and assessment of the presented work, I am conflicted in my assessment of the relevance of this work. It is in my opinion that this paper errs on the path of doing science for the sake of it, instead of being applied to a know and well-posed problem, which I don't think is a bad thing in itself.

**Questions To Address In The Rebuttal:**

Most importantly, the authors should put forward the motivations of exploring unlearning, and ideally review the aim of the article so that applications of unlearning are better put forward. Actual applications of unlearning could be explored in the experiments and results.  Please address the comments in the section above.

**Special Issue:**

No

---

> ### Author Response · Authors · 2024-03-15
> **Rebuttal to Reviewer fLbn (1)**
>
> The authors thank reviewer fLbn for the time and effort for reviewing our work and the constructive advice. Here is our reply to the comments.
>
> ---
>
> **Q1**: *Unfortunately, the task of unlearning in MRI as presented in this work feels artificial… does not reflect a real life problem… does not represent a meaningful scenario... the authors should put forward the motivations of exploring unlearning, and ideally review the aim of the article so that applications of unlearning are better put forward.*
>
> **A1**: Thank you for raising the question of the motivation. We reemphasize that our work is grounded in emerging challenges within the field of MRI analysis. A growing body of research has identified the phenomenon of hallucinations in MRI reconstructions — artifacts that can severely distort clinical interpretations [1, 2, 3]. These hallucinations are due to the instability of AI-based reconstruction methods, highlighting a critical issue that can directly impact patient care.
>
> Currently, we lack both controllable methods capable of creating realistic artifacts and further, oracle detectors capable of reliably detecting and locating artifact pre-existence. We acknowledge that our nascent experimental setup therefore leverages artificially induced artifacts that none-the-less, enable us to illustrate and explore method abilities to unlearn synthetic visual artifacts that we believe provide a useful proxy for meaningful scenarios.
>
> The reported scenario is meaningful as it approximates genuine potential sources of data bias (in this case by mixing data sources). Further, the simple synthetic nature of our setup affords future work with an easy and intuitive path to experimental reproducibility.
>
> ---
>
> **Q2**: *The considered evaluation metrics do not tell us if hallucinations are removed or if knee images can no longer be recovered, for example.*
>
> **A2**: Indeed, metrics such as mean squared error (`MSE`) and peak signal-to-noise ratio (`PSNR`) predominantly quantify image fidelity in comparison to Groundtruth. However, these metrics also indirectly reflect the absence of hallucinations; hallucinations would typically contribute to increased error or a lower `PSNR` value by introducing discrepancies between the reconstructed image and the Groundtruth (shown in Fig. 1, error calculated by `MSE`). However, we recognize the limitation of using `PSNR` alone to conclusively demonstrate the removal of hallucinations or the prevention of knee image recovery in a strict sense.
>
> In light of this, future work will look to integrate additional qualitative analyses, potentially including expert radiologist evaluations or advanced hallucination-specific detection methodologies, to supplement our quantitative metrics. Expert review is time consuming and absence of datasets that allow us to train unbiased detectors render the automated detection also difficult.
>
> ---
>
> To be continued on `Rebuttal to Reviewer fLbn (2)`.
>
>
> **Reference**:
>
> [1]: Antun, V., Renna, F., Poon, C., Adcock, B., Hansen, A.C., 2020. On instabilities of deep learning in image reconstruction and the potential costs of AI. Proc. Natl. Acad. Sci. 117, 30088–30095.
>
> [2]:  Bhadra, S., Kelkar, V.A., Brooks, F.J., Anastasio, M.A., 2021. On Hallucinations in Tomographic Image Reconstruction. IEEE Trans. Med. Imaging 40, 3249–3260.
>
> [3]:  Darestani, M.Z., Chaudhari, A.S., Heckel, R., 2021. Measuring Robustness in Deep Learning Based Compressive Sensing, in: Proceedings of the 38th International Conference on Machine Learning. Presented at the International Conference on Machine Learning, PMLR, pp. 2433–2444.

---

> ### Author Response · Authors · 2024-03-15
> **Rebuttal to Reviewer fLbn (2)**
>
> Continued from `Rebuttal to Reviewer fLbn (1)`.
>
> ---
>
> **Q3**: *Moreover, the presented results are unconvincing and none seem to outperform the others. Most considered methods either provide no impact wrt their "oracle" network, or catastrophically degrade the results on the target "retain" dataset. Therefore, even though the problem is novel, there is seemingly no grounds for a potential future avenue of research.*
>
> **A3**: We wish to clarify the multi-dimensional approach we used to evaluate the effectiveness of our unlearning method. Specifically, our evaluation framework considers: 1) accuracy on in-distribution data (brain data), 2) the degree to which model performance on out-of-distribution data (knee data) approximates oracle network performance, and 3) the run-time efficiency of the model. This allows for a nuanced understanding of the advantages and trade-offs intrinsic to unlearning. Upon reviewing _Table 1_ and _Figure 3_, we have observed that the Noisy Label Fine-Tuning (`NL-FT`) method demonstrates a balanced profile across these metrics. While `NL-FT` ranks third in Brain Test Accuracy (`BTA`), it secures the first position in Knee Test Accuracy (`KTA`) and Unlearning Accuracy (`UA`), while maintaining relatively high Run-Time Efficacy (`RTE`). This profile highlights a significant step forward in unlearning to achieve a balance across multiple critical dimensions. The improvements in unlearning offer a compelling argument for continued exploration in this field. The potential to refine these models further and adapt the unlearning techniques to more dynamically reallocate a model's representational capacity presents an exciting and viable future research avenue. Our work serves to lay the groundwork for such advancements in the understanding and application of unlearning.
>
> ---
>
> **Q4**: *Run-time efficiency (RTE) is introduced as a metric in section 4.3 but is never explained. Figure 3 displays it for some considered method but does not bring anything to a reader not familiar with the metric. Figure 3, with our without RTE, seems redundant with Table 1 and could be excluded from the article.*
>
> **A4**: In _Section 4.3_, we introduced run-time efficiency (`RTE`) as a metric to quantify the operational efficiency of our unlearning model. We will amend the paper to include a clear and comprehensive explanation of `RTE`. Run-time efficiency will be defined as *the time taken by the model to carry out an unlearning process, divided by the total time used for retraining the whole model*.
>
> ---
>
> **Q5**: *4.3 refers to Sub-FT and Full-FT yet table 1 considers FT and Full-FT.*
>
> **A5**: Thank you for pointing out the discrepancy in the terminology used in _Section 4.3_ and _Table 1_. We acknowledge this inconsistency and will amend this in the manuscript to ensure clarity and uniformity. The term `Sub-FT` mentioned in _Section 4.3_ is indeed intended to correspond to `FT` as presented in _Table 1_. Similarly, `Full-FT` in both the text and the table refers to the same fine-tuning method applied across the full dataset. We will rectify this oversight to align the narrative of _Section 4.3_, with narrative of _Table 1_, making it clear that `Sub-FT` and `FT` are synonymous.
>
> ---
>
> **Q6**: *The results in Figure 4 seem very sparse and could do with more samples.*
>
> **A6**: _Figure 4_ shows an ablation study of how many retain data are utilised to keep the unlearned model in-distribution, while the x-axis is the related unlearning time with respect to the data portion and the y-axis shows the corresponding brain test accuracy in `PSNR`. The sparse image shows that with the increase of data samples, the time required will increase significantly, as we choose some representative portion of the data. We will add more data portions as a test of unlearning efficiency.

---

> > ### Comment · Reviewer_fLbn · 2024-03-18
> >
> > Thank you for the thorough and well presented rebuttal. The authors have done a good job of addressing the concerns raised and I agree with most of the responses. I have no further comments.

---

> > > ### Author Response · Authors · 2024-03-18
> > >
> > > We sincerely appreciate reviewer fLbn's thoughtful review and the time you invested in evaluating our manuscript. We are glad to hear that our rebuttal has satisfactorily addressed your concerns, and we are grateful for your agreement with our responses!

---

### Official Review · Reviewer_aeoX · 2024-02-28

**Confidence:** 3
**Preliminary Rating:** 4

**Summary:**

The authors employ unlearning as a means to eliminate hallucinations, serving as a representative example of removing undesired data. Their findings demonstrate the feasibility of machine unlearning without the necessity for complete retraining, and they suggest that it is possible to maintain high performance levels even when utilizing only a subset of retained data.

**Strengths:**

The paper provides a thorough and examination of unlearning techniques applied to MRI reconstruction. This is new in the field. Then, this will surely have an impact and a discussion (positive or not) that may be beneficial in a congress like MIDL

**Weaknesses:**

"high performance levels" is not strictely defined
Although their mentioned it as future work, there is not much discussion on why, when the amount of retained set data increases the quality of image reconstructions does not improve
May be this due to some effect of the l1 norm in 2?
How are the model parameters?

**Detailed Comments:**

The paper is thoroughly detailed, presenting a comprehensive exploration of unlearning techniques applied to MRI reconstruction. It serves as a potential pioneering work in the domain of unlearning for MRI reconstruction.
However, some details are omitted during the methodology description in 3.3 that would improve the comprehension and reproducibility

**Justification Of The Preliminary Rating:**

Same that asked in rebuttal: The paper provides a thorough and comprehensive examination of unlearning techniques applied to MRI reconstruction. It represents a potential pioneering contribution in the field of unlearning for MRI reconstruction. However, there are certain details omitted in the methodology description in section 3.3 that, if included, would enhance comprehension and reproducibility.

**Questions To Address In The Rebuttal:**

The paper provides a thorough and comprehensive examination of unlearning techniques applied to MRI reconstruction. It represents a potential pioneering contribution in the field of unlearning for MRI reconstruction. However, there are certain details omitted in the methodology description in section 3.3 that, if included, would enhance comprehension and reproducibility.

---

> ### Author Response · Authors · 2024-03-15
> **Rebuttal to Reviewer aeoX**
>
> The authors thank reviewer aeoX for the effort for reviewing our work and the constructive suggestions. Here is the rebuttal to the questions.
>
> ---
>
> **Q1**: *"high performance levels" is not strictly defined*.
>
> **A1**: Let us first clarify what we mean by "high performance levels", which is the ability of a model to achieve high-quality reconstruction for the domain it has retained, referred to in-distribution data; in the context of our study, this refers to brain imaging data. A model should strive to maintain comparable reconstruction quality for data from outside the domain, known as out-of-distribution data, such as knee imaging data. We further aim to characterise "high performance" in the context of unlearning by setting a benchmark against what we refer to as the oracle model denoted by  $\hat{G}$. Our definition of successful unlearning is captured by a model performance to approach the oracle model’s performance on in-distribution data as well as on out-of-distribution data. This has been defined in _Section 3.2_ of our manuscript, where we provide a formal definition and elaborate on the assessment metrics. Moreover, another factor on which to measure unlearning performance is computational run-time, noted as run-time efficiency (`RTE`) in our manuscript. In the revised manuscript, we will re-define the goals to avoid any potential ambiguity.
>
> ---
>
> **Q2**: *Although their mentioned it as future work, there is not much discussion on why, when the amount of retained set data increases the quality of image reconstructions does not improve. May be this due to some effect of the l1 norm in 2? How are the model parameters?*
>
> **A2**: We observed that an increase in the amount of retained set data can improve the reconstruction performance when starting from 1%, as illustrated in _Figure 4_, and then decreases the performance from a certain percentage, which we speculate this is owing to the distribution shift between the retained set and the large-scale training set. Fitting the model to the comparatively small amount of retained data can degrade the reconstruction performance thought it may enhance the unlearning efficacy. Both `GA-L1-FT` and `NL-FT` has the same trend, while `GA-L1-FT` is constrained with $\ell$1-norm, while `NL-FT` is not. As for the model capacity, in the future work we would conduct more experiments on different scale of reconstruction model to get to know better the relationship of model size and the amount of retained set data in unlearning.
>
> ---
>
> **Q3**: *However, some details are omitted during the methodology description in 3.3.*
>
> **A3**: All details about the methodology are in the Appendix. We hope this solution offers the ideal balance  between explicit clarity and tight spatial constraints; we will happily take suggestions  to make further amendments as required.

---

> > ### Comment · Reviewer_aeoX · 2024-03-27
> > **Rebuttal**
> >
> > Thank you for the rebuttal, adding the definition in section 3.2 and add the details of the methodology in Annex I. The authors have effectively addressed the raised concerns, and I concur with the majority of the responses.

---

> > > ### Author Response · Authors · 2024-03-27
> > >
> > > We sincerely appreciate reviewer aeoX's thoughtful review and the time you invested in evaluating our manuscript. We are glad to hear that our rebuttal has satisfactorily addressed concerns you raised, and we are grateful for your concurrence.

---

### Meta-Review · Area_Chair_X59N · 2024-04-04

**Recommendation:** Accept (Poster)
**Confidence:** 5

**Metareview:**

The reviewers find the work interesting and the results promising. I concur with the reviewers that this is an interesting approach to a problem that has not been heavily explored yet and that the paper would be of interest to the MIDL community.

---

### Decision · Program_Chairs · 2024-04-06

Accept (Poster)